# Trends in health expectancy at age 60 in Bangladesh from 1996 to 2016

Md. Ismail Tareque  *

Department of Population Science and Human Resource Development, University of Rajshahi, Rajshahi, Bangladesh

* tarequemi_pops@ru.ac.bd, tareque_pshd@yahoo.com

## Abstract

### Background

Life expectancy (LE) is increasing all over the world, and relying on LE alone is no longer sufficient to identify whether a country is having a healthier population. Examining the increase in LE in relation to health — health expectancy estimation — is advised to ascertain the increase (or decrease) in LE without disability over time. This study examines the trends in health expectancy at age 60 in Bangladesh from 1996 to 2016.

### Methods

Mortality information from United Nations and World Health Organization and morbidity information from Bangladesh Bureau of Statistics were combined using the Sullivan method.

### Results

With an overall declining trend over the study period and a big drop in disability rates during 2012–2013, the disability rates were observed 1.6–1.7% in 2016. The declining trend in disability may have two-fold implications: (1) among the 98.3% older adults (≥60 years) with no severe/extreme disability, those were in jobs could have continued their work if there was no mandatory retirement at age 59, and (2) the 1.7% (translates into 0.2 million in 2020) older adults with severe/extreme disability require care assistance with their daily activities. The observed gain in disability-free life expectancy, the decrease in life expectancy with disability and its proportion allude to the compression of morbidity and healthier older adults over time.

### Conclusion

In 2020, Bangladesh had 13.2 million (i.e., 8% of the total population) older adults, which is increasing day by day. The policy makers and government are suggested to prioritize the issues of older adults, particularly disability, care needs, retirement age, and health in the light of the current study's findings. Utilizing health expectancy research is suggested to understand the combined effect of disability and mortality for considering policy changes.

**Data Availability Statement:** All data are held in public repositories. The sources are mentioned in the Supporting information (S1 Text). Nationally representative data on SRH status by age and gender access can be obtained at www.worldvaluessurvey.org.

**Funding:** This study received support from the Faculty of Science, University of Rajshahi (grant number: A-1155-5/52/RU/Science-34/19-20). There was no additional external funding received for this study. The Faculty of Science, University of Rajshahi had no role in the study design, or interpretation of the results, or preparation, or approval of the manuscript.

**Competing interests:** The author has declared that no competing interests exist.

## Introduction

Life expectancy (LE), gauged on the basis of mortality indicators alone, has been increasing all over the world gradually for last two centuries, predominantly due to decline in early and mid-life mortality. Improvements in sanitation, housing, and education, continued with the development of vaccines and antibiotics have resulted the decline in early and mid-life mortality, which was largely due to infections [1]. In some developed countries, the current continuing increase in LE is almost entirely attributable to the decline in late-life mortality, as there was little room for further reduction in early and mid-life mortality in late twentieth century [1]. Bangladesh is experiencing epidemiological and demographic transitions, where the LE at birth has also increased from 46.6 years in 1971 to 72.9 years in 2020, mainly due to decline in early life mortality [2]. However, this increase in LE may not imply a healthier population [3, 4]. Relying on mortality indicators alone is no longer sufficient to identify whether a country is having a healthier population over time; examining the increase in LE in relation to health — health expectancy estimation — is advised to ascertain the increase (or decrease) in LE without disability over time to signify a healthier (or unhealthier) population in a country [5].

Health expectancy is defined as *"the analysis of both healthy and unhealthy years of life where health can be defined along various dimensions"* [6]. As a holistic measure of population health, it combines length and quality of life into a single metric, and thus captures both quantity and quality of LE [7–9]. When computing health expectancy employing self-rated health (SRH), the terms healthy life expectancy (HLE) and unhealthy life expectancy (UHLE) conventionally refer to the LE spent in good and poor SRH, respectively [6, 10]. When computing health expectancy employing disability, the terms disability-free life expectancy (DFLE) and life expectancy with disability (DLE) conventionally refer to the LE spent without disability and with disability, respectively [6, 11, 12].

Health expectancies are increasingly employed to evaluate changes in population health status and functional dependency, particularly in respect of older adults [8, 13]. These have become very important for projecting future levels of care needs and assistance for the older adults in the developed regions of the world [14]. For example, health expectancies were utilized to identify whether there was compression or expansion of morbidity between 1986 and 2004 in Japan [5], and between 1970 and 2010 in the USA [15], whether the UK workers are expecting healthy working life till State Pension age [16], whether the UK government's Ageing Society Grand Challenge (to increase the HLE by five years by 2035) will be attained [17], whether healthy life years are reduced for obese (versus normal weight) older adults [18], for hearing and/or vision impaired (versus non-impaired) older adults [19], and for lonely (versus non-lonely) older adults in Singapore [10]. However, there is a paucity of health expectancy research in developing countries.

In Bangladesh, some research on health expectancy exists, which reported DFLE, DLE, HLE and UHLE for people of all ages and older adults aged ≥60 years as well as HLE and UHLE for people aged ≥15 years at some points in time [11, 12, 20, 21]. In addition, a study reported partial work-loss free LE and its changes between 2004 and 2007 for men aged 30–54 years [22], and a study reported LE with and without hypertension for people aged ≥35 years [23]. In Bangladesh, at age 60, despite having longer LE, women have a greater prevalence of disability and shorter DFLE than men [12]. Inequalities in LE, DFLE and DLE between rural and urban areas are reported; urban males and females, respectively, have a longer DFLE and shorter DLE both in number and proportion when compared to rural males and females [11]. HLE declines with the increase in age in Rajshahi district of Bangladesh [20]. Men expected fewer LE spent in good SRH but a much larger proportion of LE in good SRH than did women in Bangladesh in 1996 and 2002 [21]. The study on partial work-loss free LE found

improvements in the average number of work-loss days as well as in work-loss free LE among men from 2004 to 2007; workers at age 30 in 2007 expected 212 days more work-loss free LE than workers of the same age in 2004 [22]. The study on LE with and without hypertension reported that women (versus men) could expect shorter LE without hypertension at all ages, in terms of both number and proportion of years [23].

The above statistics point to the availability of the levels of health expectancy at different ages at different points in time; but there is no study that has examined the trends in health expectancy at age 60 in Bangladesh. This study, therefore, examines the trends in health expectancy, particularly the trends in both HLE/ DFLE and UHLE/ DLE at age 60 in Bangladesh from 1996 to 2016. This trend analysis will help identify the compression or expansion of morbidity (i.e., poor health/ disability) between 1996 and 2016, thus, help appraise whether Bangladesh is having healthier older adults, consequently may help redefine retirement age. This analysis will be useful in monitoring older adults' health over time as well.

## Methods

To compute health expectancy, two key pieces of information (specifically, mortality and morbidity) are needed. Mortality information, particularly age- and sex-specific mortality rates and LE can be obtained from a standard period life table. Morbidity information, particularly age- and sex-specific prevalence or proportions of SRH/ disability can be obtained from a cross-sectional survey for the same period as the standard life table. Available mortality and morbidity data for Bangladesh along with their formats and sources are discussed in detail in (S1 Text). Only analytical data are discussed below.

### Analytical data

**Mortality information.** Five-yearly abridged life tables for 1995–2000 and 2000–2005 from World Population Prospects 2019, and yearly abridged life tables from 2009 to 2016 from World Health Organization's Global Health Observatory data repository were utilized (S1 Text).

**Morbidity information.** SRH status from the World Values Survey (WVS) for 1996 and 2002, and yearly disability prevalence for 2009–2016 from the Sample Vital Registration System (SVRS) 2010, 2012–2013, Bangladesh Sample Vital Statistics (BSVS) 2014–2018 were utilized (S1 Text).

### Analysis

At first, the prevalence of poor SRH for 1996 and 2002 and prevalence of disability from 2009 to 2016 by age-group and gender were calculated using Stata/MP version 13.0 (StataCorp LP, College Station, Texas, USA). Then *annual percent change* in disability from 2009 to 2016 (and *annual percent change* in poor SRH for 1996 and 2002) are measured using the following formula:

$$Annual\ percent\ change\ \text{in disability} = \frac{D_{t_2} - D_{t_1}}{D_{t_1} \times (t_2 - t_1)} \times 100$$

Where $D_{t_1}$ and $D_{t_2}$ are disability rates at time $t_1$ and $t_2$, respectively.

*A positive* value indicates *increment* in disability, and a *negative* value indicates *decrement* in disability.

Necessary columns of the life tables and disability prevalence from 2009 to 2016 (and prevalence of poor SRH for 1996 and 2002) were combined using the Sullivan method [24] to compute health expectancy in the current study. There are the two main methods of health

expectancy computation— the Sullivan method is a prevalence-based method, and others (e.g., multistate life table method) are incidence-based methods [6, 25]. Despite having some limitations in detecting sudden change in health transition rates, the Sullivan method is found acceptable for monitoring long term trends in health expectancies [26]. The concepts, methods and materials of health expectancy can be found elsewhere [6, 25, 27]. In the current study, health expectancy at age x was computed, separately for men and women, using the following formulas:

$$\mathrm{DFLE_x}\ (or\ \mathrm{HLE_x}) = \frac{1}{l_x} \sum_{a=x}^{w} L_a(1 - \pi_a)$$

$$\mathrm{DLE_x}\ (or\ \mathrm{UHLE_x}) = \frac{1}{l_x} \sum_{a=x}^{w} L_a \pi_a$$

Where, $l_x$ refers to the number of survivors at age x; $L_a$ refers to the person-years lived for the age interval a, $\pi_a$ refers to disability prevalence from 2009 to 2016 (and prevalence of poor SRH for 1996 and 2002) in the age interval a, and w is the last age group of the life table. Details of the estimation procedure of the Sullivan method for calculating $\mathrm{DFLE_x}$ (or $\mathrm{HLE_x}$) and $\mathrm{DLE_x}$ (or $\mathrm{UHLE_x}$) are available elsewhere [28]. All health expectancy estimations for each year, separately for men and women, were performed with MS Excel.

## Ethical considerations

This study does not need ethical approval, as it uses de-identified data from secondary sources.

## Results

### Disability prevalence

The prevalence of poor SRH for 1996 and 2002 and prevalence of disability from 2009 to 2016 and their *annual percent change* at age 60 from 1996 to 2016 in Bangladesh are provided in (S1 Table). Poor SRH status is utilized as a substitute for disability measure in 1996 and 2002, as information on disability is unavailable in 1996 and 2002. Around 264 men (per 1000) and 286 women (per 1000) were with poor SRH status in 1996. Fewer older men had poor SRH in 2002 than in 1996, and no woman had poor SRH in 2002. The disability rates and their *annual percent change* by gender from 2009 to 2016 are illustrated in Figs 1 and 2 to comprehend their trends.

As can be seen from Fig 1, at age 60, men and women were found to have almost similar rate of disability across years from 2009 to 2016 (Fig 1). Although disability has an overall declining trend from 2009 to 2016, there was a small increment in disability from 2009–2010, 2011–2012, and 2015–2016, a small decrement in disability from 2010–2011, and a large decrement from 2012–2013. Apparently, the disability rate remained almost stagnant (46–55 per 1000 older adults) from 2009 to 2012, then it decreased sharply to 14–17 (per 1000 older adults) in 2013. From 2013 to 2016, the disability rate remained almost stagnant (14–17 per 1000 older adults) again.

At age 60, for men and women respectively, there were 12.4% and 3.2% increment in disability in 2010, 16.5% and 12.5% decrement in disability in 2011, again 2.8% and 0.7% increment in 2012 (Fig 2). The largest decrements (69.8% for men and 62.5% for women) were observed in 2013. Although there were increment in disability among men and decrement in disability among women in 2014–2015, decrement among men and increment among women were found in 2016.

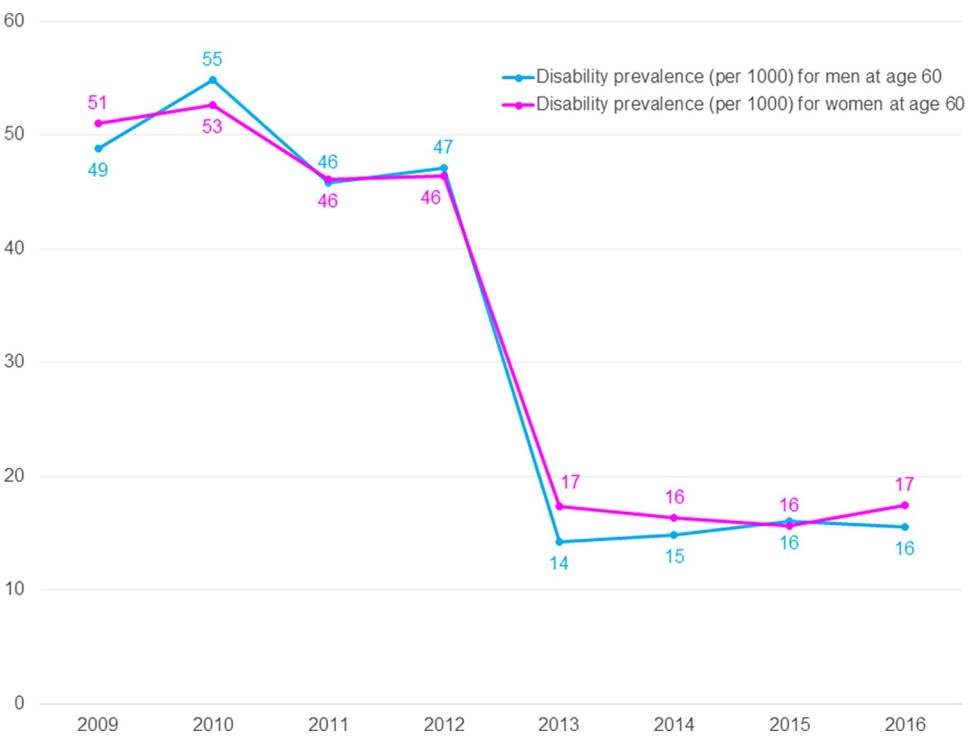

**Fig 1. Disability prevalence (per 1000) by gender at age 60 from 2009 to 2016.**

## Health expectancy

Health expectancy, particularly DFLE and DLE, for men and women separately at age 60 from 1996 to 2016 is provided in Fig 3. For 1996 and 2002, DFLE represents HLE (i.e., LE with good SRH), and DLE represents UHLE (i.e., LE with poor SRH). In 1996, at age 60 an older man could expect to live 17.25 years, of which 12.70 years with good SRH and 4.55 years (26.38% of LE) with poor SRH, and an older woman could expect to live 17.85 years, of which 12.75 years with good SRH and 5.10 years (28.57% of LE) with poor SRH. In 2002, for both men and women, there were an increase in LE, an increase in LE with good SRH, a decrease in LE with poor SRH, and a decrease in proportion of LE with poor SRH (Fig 3).

For both men and women, an increase in LE, an increase in DFLE, a decrease in DLE, and a decrease in proportion of LE with disability, with a few exceptions, across years from 2009 to 2016 were observed. Among men at age 60, there were 0.55 year increase in LE, 0.83 year increase in DFLE, and 0.28 year decrease in DLE in 7 years from 2009 to 2016 (See Panel-A, Fig 3). In 2009, an older man at age 60 could expect to live 18.01 years, of which 17.13 years without disability and 0.88 year (4.89% of LE) with disability. In 2016, an older man at age 60 could expect to live 18.56 years, of which 17.96 years without disability and 0.60 year (3.23% of LE) with disability.

Among women at age 60, there were 2 year increase in LE, 2.88 year increase in DFLE, and 0.88 year decrease in DLE in 7 years from 2009 to 2016 (See Panel-B, Fig 3). In 2009, an older woman at age 60 could expect to live 18.88 years, of which 17.92 years without disability and 0.96 year (5.08% of LE) with disability. In 2016, an older woman at age 60 could expect to live 20.88 years, of which 20.80 years without disability and 0.08 year (0.38% of LE) with disability.

Compared to men, women had longer LE and longer DFLE across years from 2009 to 2016, longer DLE from 2009 to 2012 but shorter DLE from 2013 to 2016 (Fig 4). Compared to men,

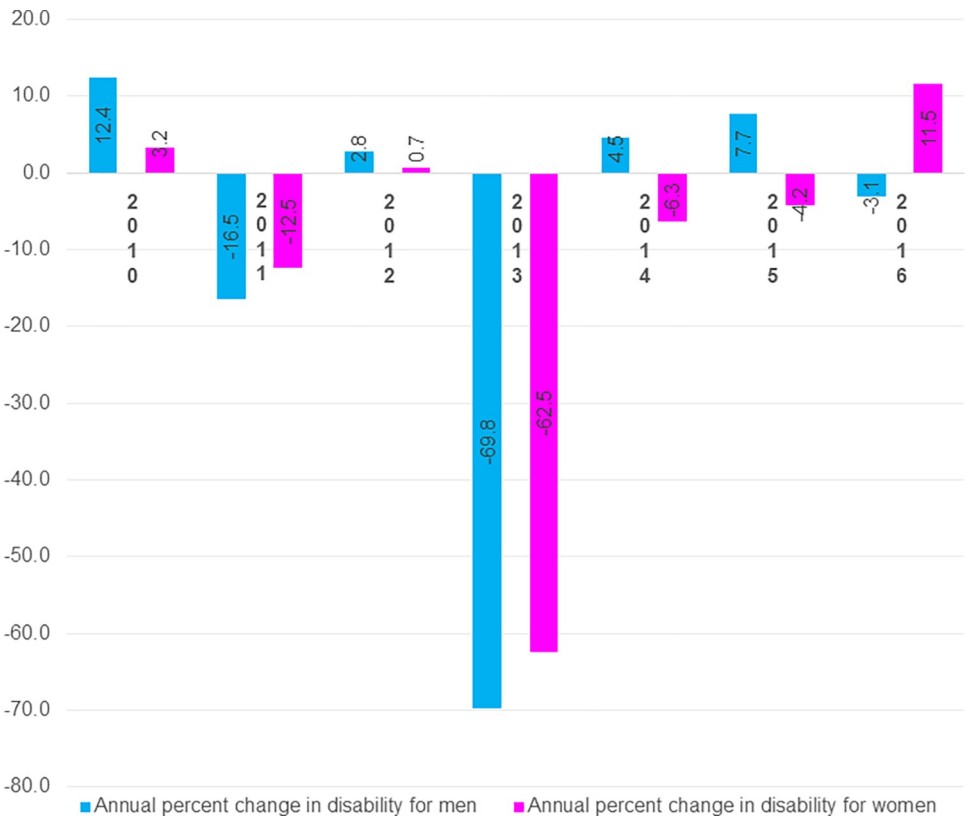

**Fig 2.** *Annual percent change* in disability by gender at age 60 from 2010 to 2016.

women had similar proportion of LE with disability from 2009 to 2012, but significantly shorter proportion of LE with disability from 2013 to 2016.

## Discussion

To my knowledge, this is the first study on the trends in health expectancy in Bangladesh. Using data from secondary sources, this study reveals some important insights, which have significant implications for disability, care needs, health expectancy, retirement and healthy aging in Bangladesh. Along with assessing poor SRH prevalence, HLE and UHLE in 1996 and 2002, the current study predominantly focuses on the trends in disability, DFLE, DLE, and proportion of LE with disability at age 60 from 2009 to 2016. With an overall declining trend over the study period and a big drop in disability rates during 2012–2013, the disability rates were observed almost stagnant at 46–55 (per 1000 older adults) during 2009–2012 and at 14–17 (per 1000 older adults) during 2013–2016. In 7 years from 2009 to 2016, at age 60, for men and women, respectively, there were 0.55 and 2.00 years increase in LE, 0.83 and 2.88 years increase in DFLE, and 0.28 and 0.88 year decrease in DLE. The decrease in the proportion of LE with disability was 1.65 percentage point for men and 4.70 percentage point for women from 2009 to 2016. Compared to men, women had longer LE and longer DFLE across years from 2009 to 2016, longer DLE (translating into similar proportion of LE with disability) from 2009 to 2012, but shorter DLE (translating into significantly shorter proportion of LE with disability) from 2013 to 2016.

Bangladesh, with a mandatory retirement age of 59 in most government and private job sectors [29, 30], defines old-age as ≥60 years in the *National Policy on Older Persons 2013*. The

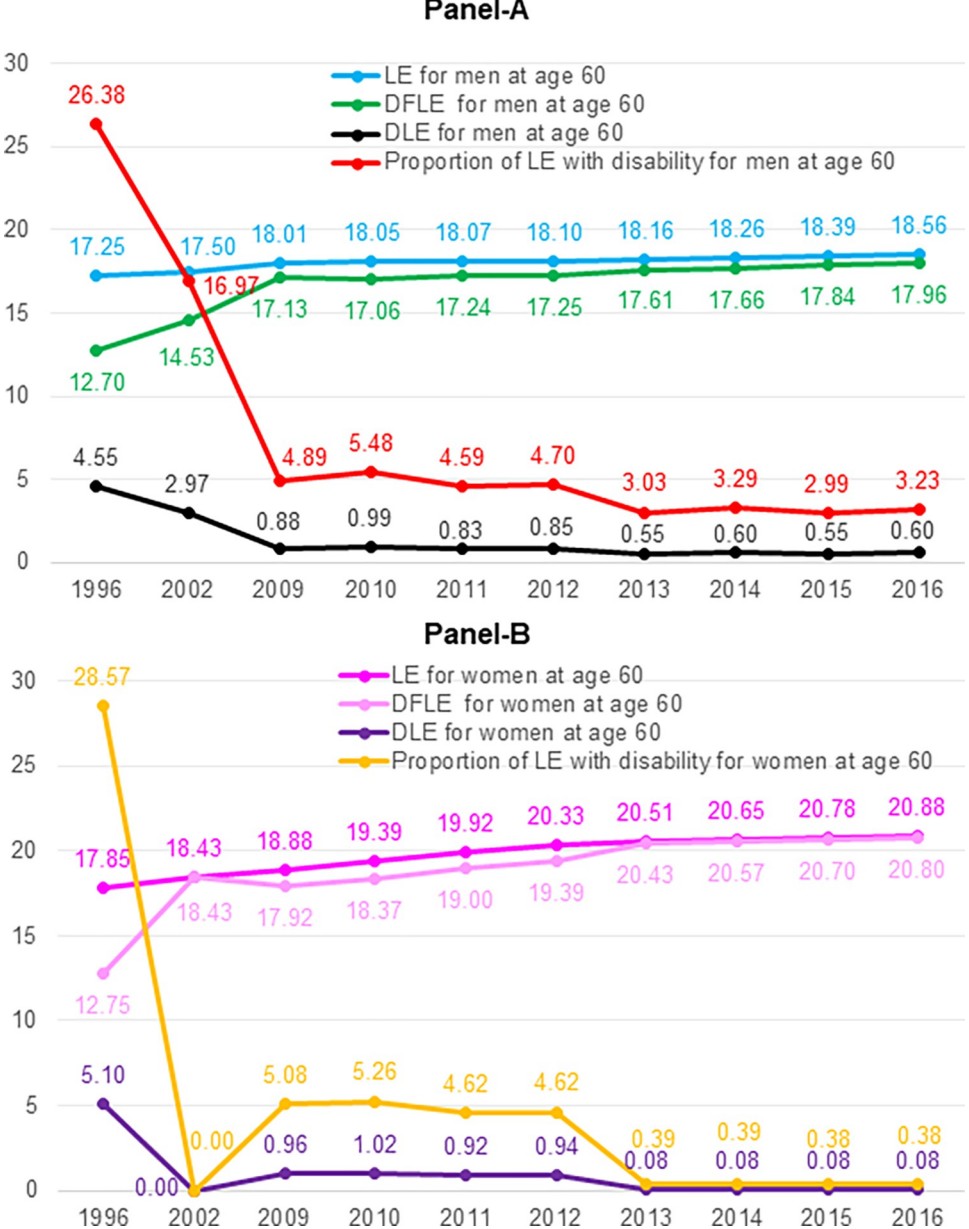

**Fig 3. Health expectancy by gender at age 60 from 1996 to 2016[†]. Notes:** [†] For 1996 and 2002, DFLE represents life expectancy with good self-reported health; DLE represents life expectancy with poor self-reported health; DFLE: Disability-free life expectancy; DLE: Life expectancy with disability; LE: Life expectancy.

disability rate coupled with the proportion of LE with disability at age 60 from the current study thus bear significant implications for redefining retirement age in Bangladesh. The disability rates observed in the current study (i.e., 4.6–5.5% during 2009–2012, reported in SVRS 2010 and 2012) are lower than that of HIES 2010, mainly due to using different approaches for measuring disability [12, 31, 32]. For instance, 42% of older adults had some form of functional disability, including 5% of older adults with severe/extreme functional disability, and 7% of older adults had a self-care disability, including 3% of older adults with a severe/extreme form of self-care disability in 2010 [32]. In 2010, 'some disability' were prevalent among 28.4% men

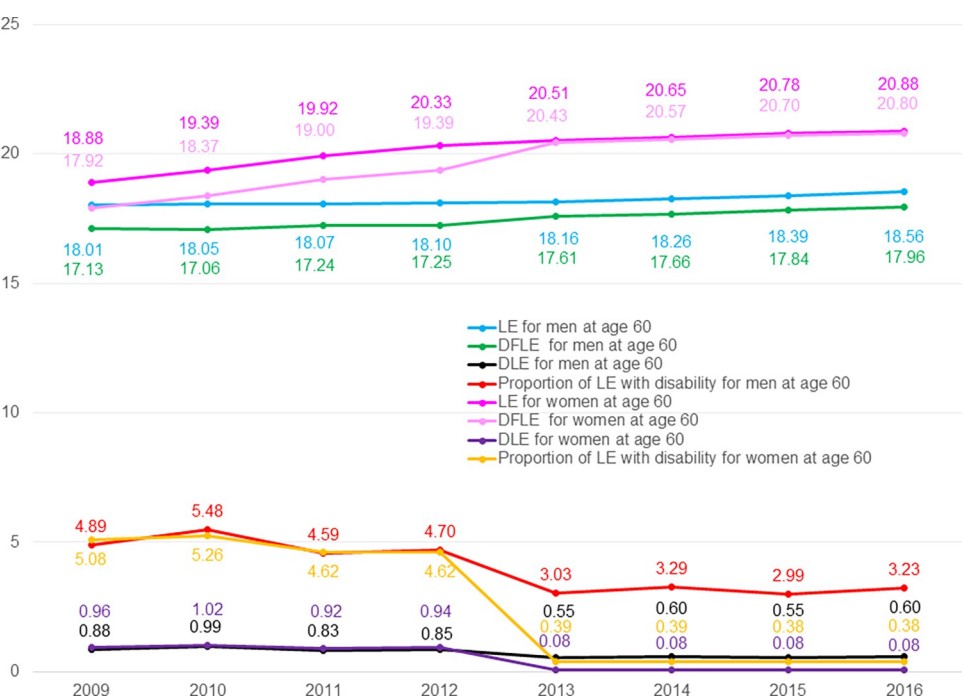

**Fig 4. Trends in health expectancy at age 60 from 2009 to 2016. Notes:** DFLE: Disability-free life expectancy; DLE: Life expectancy with disability; LE: Life expectancy.

and 34.8% women in the age group of 60–64 years, 32.2% men and 40.4% women in the age group of 65–69 years, 43.9% men and 52.1% women in the age group of 70–74 years, and so on [12]. Although, the same indicators — Washington Group's short set of six questions — were adopted in all of SVRS 2010, 2012–2013, BSVS 2014–2016, and HIES 2010, the definition of disability — how the disability was measured from the six indicators — is not explicitly stated in SVRS 2010, 2012–2013, and BSVS 2014–2016. Different cut-offs from the same Washington Group's short set of six questions produce different estimates [11]. Apparently, the observed disability rates in the current study are close to those with severely/extremely disabled in HIES 2010 [31], signifying that the severe/extreme form of disability in any of Washington Group's short set of six questions was considered as disability in SVRS 2010, 2012–2013, and BSVS 2014–2016. This conservative cut-off (i.e., severe/extreme disability in any of the six tasks) has not been utilized to define disability in the extant literature in Bangladesh [11, 12] and in other settings [33–35]. Further, such conservative definition of disability minimizes specificity, such that those designated as *'without disability'* may have 'some difficulty' with two or more of the Washington Group's recommended six tasks, and may not be able to perform a comprehensive range of activities that are required for independent living, particularly at ≥60 years. Thus, instead of conservative cut-off, adopting prudent and practicable cut-off for measuring disability, particularly for older adults, is warranted. Practicable cut-off may contemplate an individual as disabled if s/he is having 'severe difficulty' with any of the six indicators, or being unable ('extreme difficulty') to perform any of the six indicators at all, or having 'some difficulty' with at least two of the six indicators [11, 33–35]. In addition, an explicit definition with cut-offs to measure disability from the Washington Group's short set of six questions in upcoming government reports (e.g., BSVS) is vital.

To my knowledge, no intervention was made during 2012–2013 or earlier that could reduce disability rate among older adults from 2012 to 2013. However, the declining trend in

disability indicates only 1.6–1.7% of older adults were disabled at age ≥60 in 2016, leaving more than 98.3% of older adults with no severe/extreme disability for functional and self-care activities. In 2020, 1.7% of Bangladesh older adults translates into 0.2 million people of age ≥60 years [36]. The declining trend in disability, with an indication of compression of morbidity, may have two-fold implications: (1) among the 98.3% older adults with no severe/extreme disability, those were in jobs could have continued their work if there was no mandatory retirement at age 59, and (2) the 1.7% older adults with severe/extreme disability require care assistance with their daily activities, and thus monitoring is suggested whether those older adults receive necessary care and support. Policy makers and government should place emphasis on this two-fold implications to make the older adults healthy and able to do their daily activities and work successfully.

In 7 years from 2009 to 2016, for men and women, respectively, there were 0.55 and 2.00 years increase in LE. Declining neonatal, infant, child and maternal mortality are the main drivers for these increases in LE. The gain in LE is lower for men than women. The causes of death that most contribute to a lower LE for men than women are ischaemic heart disease, road injuries, lung cancer, chronic obstructive pulmonary disease, stroke, cirrhosis of the liver, tuberculosis, prostate cancer [37]. Declining maternal mortality may also contribute to extra gain in LE for women than men. In 7 years from 2009 to 2016, the gain in DFLE is greater than the gain in LE, and there was a decrease in DLE. The gain in DFLE, the decrease in DLE and the decrease in the proportion of LE with disability allude strongly to the compression of morbidity during 2009–2016. These also indicate that Bangladesh is having healthier older adults over time. The primary reason for the gain in DFLE is the decline in disability rates over time. In addition, medical advances may be reducing the likelihood that diseases progress to disabling conditions [38]. Maintaining health and reducing disability at younger ages are suggested to have meaningful compression of morbidity at older ages as well as healthier older adults [15]. Behavioral risk factors (smoking, physical inactivity, obesity, alcohol consumption, and unhealthy lifestyle) [18, 39–42], hearing and vision impairments [19], sleep disturbances [43], and loneliness [10] are found to be contributors to shorter LE and shorter HLE/DFLE in different settings. These factors may be taken into consideration to further improve the DFLE among older adults in Bangladesh. The current study mainly demonstrates the value of utilizing health expectancy research to understand the combined effect of trends in disability and mortality for considering policy changes (e.g., definition of older adults, retirement age, prioritizing health expectancy research and healthy aging).

Women (versus men) had longer LE, longer DFLE across years from 2009 to 2016, longer DLE (i.e., similar proportion of LE with disability) from 2009 to 2012, but shorter DLE (i.e., significantly shorter proportion of LE with disability) from 2013 to 2016. Due to having similar rate of disability among men and women in the current study, longer LE and longer DFLE among women than men may be attributable to survival and biological sex differences [37]. Longer LE and longer DFLE among women than men in the current study are contrary to the findings that women have shorter DFLE than men, in numbers and proportion [12]. Using different cut-offs for disability measurement in the earlier study [12] and the current ones are the possible reasons. More investigation on the gender differences in the trends in health expectancy is indispensable.

I acknowledge a few limitations of this study. The self-reported disability was utilized in the current study that could be a possible source of bias, though the self-reported functional disability was found to be consistent with medical diagnoses [44]. The SRH was also utilized as a measure of morbidity, which may have introduced gender bias in the findings [45]. The single-item SRH was however reported to be able to capture almost the same as objective health status in predicting a short-term mortality among older adults [46]. Due to unavailability of

same morbidity measure across the study period, two different subjective morbidity measures — SRH and disability — were utilized in the current study. They both were found to be consistent with their objective measures, and were reported to predict mortality among older adults in different settings [44, 46, 47]. Institutionalized populations were not interviewed in SVRS 2010, 2012–2013, and BSVS 2014–2016, but life tables were computed based on the mortality information of all institutionalized and non-institutionalized (i.e., community people). The DFLE might either be over-estimated if institutionalized populations had higher disability rates than community people, or under-estimated if the institutionalized populations had lower disability than community people. The same disability rate was assumed for both the institutionalized and non-institutionalized populations in the current study. Some older adults might have been unable to respond to the interviewers, and some proxy respondents instead might have been interviewed in SVRS 2010, 2012–2013, and BSVS 2014–2016. The SVRS and BSVS stated nothing about proxy respondents. Disability rates for the year 2017 and 2018 are available in BSVS 2017 and 2018, but are not used for this study due to unavailability of yearly life table for the year 2017 and 2018 in the World Health Organization's Global Health Observatory data repository. LE at birth in 2017 (Male: 70.6 years; Female: 73.5 years) and 2018 (Male: 70.8 years; Female: 73.8 years) in Reports on Bangladesh Sample Vital Statistics 2017 and 2018 are lower than that in 2016 (Male: 71.1 years; Female: 74.4 years) in World Health Organization's Global Health Observatory data repository. Inclusion of life tables for 2017 and 2018 from Reports on Bangladesh Sample Vital Statistics may not represent the proper trends in LE; thus, trends in health expectancy was not extended to 2017 or 2018.

## Conclusion

Bangladesh is not yet considered an 'aging society,' but in 2020, it had 13.2 million (i.e., 8% of the total population) people aged ≥60 years, which is more than the total population of some countries (e.g., in 2020, Singapore's total population is 5.9 million, and China, Hong Kong SAR's total population is 7.5 million [36]). The number of the older adults is increasing day by day. The policy makers and government are suggested to prioritize the problems and issues of the older adults, particularly disability, care needs, retirement age, and health in the light of the current study's findings. Utilizing health expectancy research is suggested to understand the combined effect of disability and mortality for considering policy changes for older adults in Bangladesh.

## Supporting information

**S1 Text. Available data.**
(DOCX)

**S1 Table. Poor SRH/ disability prevalence (per 1000) and *annual percent change* in poor SRH/ disability by gender at age 60 from 1996 to 2016.**
(DOCX)

## Author Contributions

**Conceptualization:** Md. Ismail Tareque.

**Data curation:** Md. Ismail Tareque.

**Formal analysis:** Md. Ismail Tareque.

**Funding acquisition:** Md. Ismail Tareque.

**Investigation:** Md. Ismail Tareque.

**Methodology:** Md. Ismail Tareque.

**Validation:** Md. Ismail Tareque.

**Visualization:** Md. Ismail Tareque.

**Writing – original draft:** Md. Ismail Tareque.

**Writing – review & editing:** Md. Ismail Tareque.

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
