## [Decision Letter · Decision Letter 0]

20 Sep 2022

PONE-D-22-03689Trends in health expectancy at age 60 in Bangladesh from 1996 to 2016PLOS ONE

Dear Dr. Tareque,

Thank you for submitting your manuscript to PLOS ONE. After careful consideration, we feel that it has merit but does not fully meet PLOS ONE’s publication criteria as it currently stands. Therefore, we invite you to submit a revised version of the manuscript that addresses the points raised during the review process.

We look forward to receiving your revised manuscript.

Kind regards,

Allen Prabhaker Ugargol

Academic Editor

PLOS ONE

Journal Requirements:

2. PLOS ONE does not copy edit accepted manuscripts (https://journals.plos.org/plosone/s/criteria-for-publication#loc-5). To that effect, please ensure that your submission is free of typos and grammatical errors.

“This study received partial support from the Faculty of Science, University of Rajshahi (grant number: A-1155-5/52/RU/Science-34/19-20). The funders had no role in study design, data collection and analysis, decision to publish, or preparation of the manuscript.”

“This study received partial support from the Faculty of Science, University of Rajshahi (grant number: A-1155-5/52/RU/Science-34/19-20). The Faculty of Science, University of Rajshahi had no role in the study design, or interpretation of the results, or preparation, or approval of the manuscript.”

“This study received partial support from the Faculty of Science, University of Rajshahi (grant number: A-1155-5/52/RU/Science-34/19-20). The funders had no role in study design, data collection and analysis, decision to publish, or preparation of the manuscript.”

Reviewers' comments:

Reviewer's Responses to Questions

**Comments to the Author**

1. Is the manuscript technically sound, and do the data support the conclusions?

Reviewer #1: Yes

Reviewer #2: Partly

2. Has the statistical analysis been performed appropriately and rigorously? 

Reviewer #1: Yes

Reviewer #2: Yes

3. Have the authors made all data underlying the findings in their manuscript fully available?

Reviewer #1: Yes

Reviewer #2: Yes

4. Is the manuscript presented in an intelligible fashion and written in standard English?

Reviewer #1: Yes

Reviewer #2: Yes

5. Review Comments to the Author

Reviewer #1: The study is a very novel attempt in the context of Bangladesh. The study has estimated and examined the trends in "Health expectancy", one of the new and intuitive measure of understanding health status of a population. As research in the context of Bangladesh has been largely concentrated around maternal and child health, this timely piece will provide a foresight for the country's policymaker to tackle ageing related health-issues in coming years.

The paper has articulated its findings very clearly along with cited all previously published major research works in this area. The paper well-written and show a decent presentation of the results. However, I would like to mention few points which can be included in the work if the author feels so.

1. A short discussion on Sullivan's method including its limitations (will be useful for readers new in the field).

2. The author may consider to include the supplementary figures 1 and 2 to the main document.

Reviewer #2: The authors attempt to provide essential insights on disability free life expectancy and correctly summarize its necessity vis-à-vis just LE. It is a timely assessment given the ageing population in Bangladesh.

Major concerns:

1. The authors provide information on various data sources available for disability and life tables- can be shifted to supplementary . They must mention and with details only the data sets used in main manuscript.

2. Since SRH and disability rates are both used, the authors should provide more information on the pros and cons of these- and justify a comparison across time with two different measures.

3. The introduction should focus more on literature about SRH and disability instead of various uses of disability free life expectancy

Minor suggestions:

1. Abstract- implications should be discussed in text and mention only major findings here

2. Introduction- line 94: can be just "or USA" instead of repeating entire sentence again

6. PLOS authors have the option to publish the peer review history of their article (what does this mean?). If published, this will include your full peer review and any attached files.

Reviewer #1: No

Reviewer #2: No

---

## [Author Response · Author response to Decision Letter 0]

9 Oct 2022

Journal Requirements:

Comment: 1. Please ensure that your manuscript meets PLOS ONE's style requirements, including those for file naming. The PLOS ONE style templates can be found at

Response: I have formatted the manuscript in accordance with PLOS ONE’s style requirements.

Comment: 2. PLOS ONE does not copy edit accepted manuscripts (https://journals.plos.org/plosone/s/criteria-for-publication#loc-5). To that effect, please ensure that your submission is free of typos and grammatical errors.

Response: I have revised the manuscript to keep it free from typos and grammatical errors. 

Comment: 3. Thank you for stating in your Funding Statement: “This study received partial support from the Faculty of Science, University of Rajshahi (grant number: A-1155-5/52/RU/Science-34/19-20). The funders had no role in study design, data collection and analysis, decision to publish, or preparation of the manuscript.”

Response: The study received only one small funding from the Faculty of Science, University of Rajshahi. I have removed “partial” from the statement, and included “There was no additional external funding received for this study.” to it. The following is the Amended Funding Statement, which is also provided in the cover letter.

“This study received support from the Faculty of Science, University of Rajshahi (grant number: A-1155-5/52/RU/Science-34/19-20). There was no additional external funding received for this study. The Faculty of Science, University of Rajshahi had no role in the study design, or interpretation of the results, or preparation, or approval of the manuscript.”

Comment: 4. Thank you for stating the following in the Acknowledgments Section of your manuscript: “This study received partial support from the Faculty of Science, University of Rajshahi (grant number: A-1155-5/52/RU/Science-34/19-20). The Faculty of Science, University of Rajshahi had no role in the study design, or interpretation of the results, or preparation, or approval of the manuscript.”

Please remove any funding-related text from the manuscript and let us know how you would like to update your Funding Statement. Currently, your Funding Statement reads as follows: “This study received partial support from the Faculty of Science, University of Rajshahi (grant number: A-1155-5/52/RU/Science-34/19-20). The funders had no role in study design, data collection and analysis, decision to publish, or preparation of the manuscript.”

Response: I have removed funding-related text from the manuscript. I have also amended the Funding statement, and provided the amended statement in the cover letter.

Comment: 5. Please include captions for your Supporting Information files at the end of your manuscript, and update any in-text citations to match accordingly. Please see our Supporting Information guidelines for more information: http://journals.plos.org/plosone/s/supporting-information.

Response: I have included the caption for the Supporting Information file at the end of the revised manuscript, and updated the in-text citations to match accordingly.

Comment: 6. Please review your reference list to ensure that it is complete and correct. If you have cited papers that have been retracted, please include the rationale for doing so in the manuscript text, or remove these references and replace them with relevant current references. Any changes to the reference list should be mentioned in the rebuttal letter that accompanies your revised manuscript. If you need to cite a retracted article, indicate the article’s retracted status in the References list and also include a citation and full reference for the retraction notice.

Response: I have revised the list. Newly added references are highlighted in yellow.

 

Review Comments to the Author

Reviewer #1: 

Comment: The study is a very novel attempt in the context of Bangladesh. The study has estimated and examined the trends in "Health expectancy", one of the new and intuitive measure of understanding health status of a population. As research in the context of Bangladesh has been largely concentrated around maternal and child health, this timely piece will provide a foresight for the country's policymaker to tackle ageing related health-issues in coming years.

The paper has articulated its findings very clearly along with cited all previously published major research works in this area. The paper well-written and show a decent presentation of the results. However, I would like to mention few points which can be included in the work if the author feels so.

Response: Thank you very much for your positive comments.

Comment: 1. A short discussion on Sullivan's method including its limitations (will be useful for readers new in the field).

Response: I appreciate your observation. To aid the new researchers in the field of health expectancy, the following is added to the Analysis section (lines# 158-164): 

“…There are the two main methods of health expectancy computation─ the Sullivan method is a prevalence-based method, and others (e.g., multistate life table method) are incidence-based methods [6,25]. Despite having some limitations in detecting sudden change in health transition rates, the Sullivan method is found acceptable for monitoring long term trends in health expectancies [26]. The concepts, methods and materials of health expectancy can be found elsewhere [6,25,27]. In the current study,…”

6. Saito Y, Robine J-M, Crimmins EM. The methods and materials of health expectancy. Stat J IAOS. 2014;30: 209–223. doi:10.3233/SJI-140840

25. Mathers CD. Health expectancies: an overview and critical appraisal. Summary measures of population health: concepts, ethics, measurement and applications. 2002; 177–204. 

26. Mathers CD, Robine JM. How good is Sullivan’s method for monitoring changes in population health expectancies? J Epidemiol Community Health. 1997;51: 80–86. doi:10.1136/jech.51.1.80

27. Robine J-M, Jagger C, Mathers CD, Crimmins EM, Suzman RM. Determining health expectancies. John Wiley & Sons; 2002.

Comment: 2. The author may consider to include the supplementary figures 1 and 2 to the main document.

Response: As suggested, I have included the supplementary figures 1 (currently Fig 3) and 2 (currently Fig 4) to the main document. Fig 3 of the earlier version of the manuscript is replaced with the supplementary figure 1 (currently Fig 3). The current Fig 3 displays health expectancy from 1996 to 2016 whereas the previous Fig 3 displayed health expectancy from 2009 to 2016.

Thank you very much for your consideration. 

Reviewer #2: 

Comment: The authors attempt to provide essential insights on disability free life expectancy and correctly summarize its necessity vis-à-vis just LE. It is a timely assessment given the ageing population in Bangladesh.

Response: Thank you very much for your positive comments.

Major concerns:

Comment: 1. The authors provide information on various data sources available for disability and life tables- can be shifted to supplementary . They must mention and with details only the data sets used in main manuscript.

Response: As suggested, I have moved the entire ‘Available data’ section to supplementary text (S1 Text). The WVS, SVRS and BSVS are elaborated at their first appearance in ‘Analytical data’ section. I have also stated the following in Methods section (lines# 133-135) so that the readers can locate the ‘Available data’ section from the supplementary files.

“……Available mortality and morbidity data for Bangladesh along with their formats and sources are discussed in detail in supplementary text (S1 Text). Only analytical data are discussed below.”

Comment: 2. Since SRH and disability rates are both used, the authors should provide more information on the pros and cons of these- and justify a comparison across time with two different measures.

Response: I appreciate your observation. I did include the limitation of using self-reported disability in the earlier version of the manuscript, but I missed to add the bias that might have been introduced by using SRH as a measure of morbidity. I have added the following to the limitation section (lines# 345-352): 

“…The SRH was also utilized as a measure of morbidity, which may have introduced gender bias in the findings [45]. The single-item SRH was however reported to be able to capture almost the same as objective health status in predicting a short-term mortality among older adults [46]. Due to unavailability of same morbidity measure across the study period, two different subjective morbidity measures ─ SRH and disability ─ were utilized in the current study. They both were found to be consistent with their objective measures, and were reported to predict mortality among older adults in different settings [44,46,47].”

44. Zunzunegui M-V, Alvarado B-E, Béland F, Vissandjee B. Explaining health differences between men and women in later life: A cross-city comparison in Latin America and the Caribbean. Soc Sci Med. 2009;68: 235–242. doi:10.1016/j.socscimed.2008.10.031

45. Case A, Paxson C. Sex differences in morbidity and mortality. Demography. 2005;42: 189–214. doi:10.1353/dem.2005.0011

46. Wuorela M, Lavonius S, Salminen M, Vahlberg T, Viitanen M, Viikari L. Self-rated health and objective health status as predictors of all-cause mortality among older people: a prospective study with a 5-, 10-, and 27-year follow-up. BMC Geriatr. 2020;20: 120. doi:10.1186/s12877-020-01516-9

47. Gobbens RJ, van der Ploeg T. The prediction of mortality by disability among Dutch community-dwelling older people. Clin Interv Aging. 2020;15: 1897–1906. doi:10.2147/CIA.S271800

Comment: 3. The introduction should focus more on literature about SRH and disability instead of various uses of disability free life expectancy

Response: The main focus of this paper is health expectancy. Therefore, life expectancy, health expectancy, and their literature review are articulated in Introduction section. The SRH and disability, mainly because of availability and employability, were utilized as the measures of morbidity, which were combined with mortality information for computing health expectancy. Different morbidity measures could be used if available and employable. The SRH and disability are not the main focus of this study, and the study’s focus and succinctness will be perished if I focus on literature about SRH and disability instead of health expectancy. In addition, I have not found a suitable place in Introduction section, where the literature about SRH and disability can be accommodated.

My response to your comment#2 is partly applicable to respond to this comment. For example, I have added the limitation of using SRH and disability along with their predictability of mortality. The definitions of SRH and disability were in the main document in the earlier version of the manuscript. Now they are shifted to the supplementary text in the light of your suggestion and comment#1. 

Minor suggestions:

Comment: 1. Abstract- implications should be discussed in text and mention only major findings here

Response: The following is found ignorable, and has been removed from the first sentence of the Results section of the Abstract. The Results section is revised, and other sections remain same.

“…almost stagnant at 4.6-5.5% during 2009-2012 and at…”

The first sentence is revised as follows:

“With an overall declining trend over the study period and a big drop in disability rates during 2012-2013, the disability rates were observed 1.6-1.7% in 2016.”

Comment: 2. Introduction- line 94: can be just "or USA" instead of repeating entire sentence again

Response: I have eliminated the repetition. I have included the time period for the USA study, and revised the sentence as follows:

“For example, health expectancies were utilized to identify whether there was compression or expansion of morbidity between 1986 and 2004 in Japan [5], and between 1970 and 2010 in the USA [15], whether……” 

Thank you very much for your consideration.

---

## [Editor Report · Decision Letter 1]

10 Nov 2022

Trends in health expectancy at age 60 in Bangladesh from 1996 to 2016

PONE-D-22-03689R1

Dear Dr. Md. Ismail Tareque,

We’re pleased to inform you that your manuscript has been judged scientifically suitable for publication and will be formally accepted for publication once it meets all outstanding technical requirements.

Kind regards,

Allen Prabhaker Ugargol

Academic Editor

PLOS ONE
---

## [Editor Report · Acceptance letter]

14 Nov 2022

PONE-D-22-03689R1 

Trends in health expectancy at age 60 in Bangladesh from 1996 to 2016 

Dear Dr. Tareque:

I'm pleased to inform you that your manuscript has been deemed suitable for publication in PLOS ONE. Congratulations! Your manuscript is now with our production department. 

Kind regards, 

on behalf of

Dr. Allen Prabhaker Ugargol 

Academic Editor

PLOS ONE